# Novel Acetylcholinesterase Inhibitors Based on Uracil Moiety for Possible Treatment of Alzheimer Disease

**DOI:** 10.3390/molecules25184191

**Published:** 2020-09-12

**Authors:** Vyacheslav E. Semenov, Irina V. Zueva, Marat A. Mukhamedyarov, Sofya V. Lushchekina, Elena O. Petukhova, Lilya M. Gubaidullina, Evgeniya S. Krylova, Lilya F. Saifina, Oksana A. Lenina, Konstantin A. Petrov

**Affiliations:** 1Arbuzov Institute of Organic and Physical Chemistry, FRC Kazan Scientific Center of RAS, Arbuzov str. 8, 420088 Kazan, Russia; zueva.irina.vladimirovna@gmail.com (I.V.Z.); salyakhova.lm@mail.ru (L.M.G.); evgeniya.s.krylova@yandex.ru (E.S.K.); dimple@mail.ru (L.F.S.); leninaox@mail.ru (O.A.L.); 2Institute of Neuroscience, Kazan State Medical University, 420012 Kazan, Russia; maratm80@list.ru (M.A.M.); petukhovaeo@mail.ru (E.O.P.); 3Emanuel Institute of Biochemical Physics, Kosygina st. 4, 119334 Moscow, Russia; sofya.lushchekina@gmail.com; 4Institute of Fundamental Medicine and Biology, Kazan Federal University, Kremlyovskaya str., 18, 420008 Kazan, Russia

**Keywords:** acetylcholinesterase, 6-methyluracil, inhibitors, peripheral anionic site, Alzheimer disease

## Abstract

In this study, novel derivatives based on 6-methyluracil and condensed uracil were synthesized, namely, 2,4-quinazoline-2,4-dione with ω-(*ortho*-nitrilebenzylethylamino) alkyl chains at the N atoms of the pyrimidine ring. In this series of synthesized compounds, the polymethylene chains were varied from having tetra- to hexamethylene chains, and secondary NH, tertiary ethylamino, and quaternary ammonium groups were introduced into the chains. The molecular modeling of the compounds indicated that they could function as dual binding site acetylcholinesterase inhibitors, binding to both the peripheral anionic site and active site. The data from in vitro experiments show that the most active compounds exhibit affinity toward acetylcholinesterase within a nanomolar range, with selectivity for acetylcholinesterase over butyrylcholinesterase reaching four orders of magnitude. In vivo biological assays demonstrated the potency of these compounds in the treatment of memory impairment using an animal model of Alzheimer disease.

## 1. Introduction

Alzheimer disease (AD) is the most common neurodegenerative disorder spread worldwide and is estimated to affect around 90 million people by 2050 [1]. AD is characterized by progressive memory loss, decline in language skills and other cognitive impairments, and severe behavioral abnormalities, ultimately resulting in death [2,3,4]. The brain of an AD patient is characterized by a loss of cholinergic neurons and a decreased number of synapses in specific areas, including the hippocampus, basal forebrain, and cortex, which are involved in learning and memory [5].

The pathogenesis of AD is not yet completely understood, with multiple factors all contributing to neuronal cell death to different extents [6,7,8]. Such factors may, for example, be lifestyle factors and conditions leading to upregulated production of proinflammatory cytokines [9,10]. Another factor is the accumulation of abnormal deposits of β-amyloid peptide (Aβ) and hyperphosphorylated tau protein in addition to widespread cell death and the loss of synapses, especially in the cholinergic system, which are considered to define the pathophysiology of AD [11]. Aβ possesses a broad range of neurotoxic effects, including oxidative stress, mitochondrial dysfunction, impairment of ion transport, synaptic dysfunction, and neuronal apoptosis [12,13]. The reduction of Aβ production and increased clearance of pathogenic Aβ forms are some of the key targets in the development of oncoming therapeutic agents for AD treatment. Currently, the primary therapeutic treatment for AD is a cholinergic replacement strategy using acetylcholinesterase (AChE) inhibitors, namely donepezil, rivastigmine, and galantamine [14]. These drugs, able to improve memory and cognitive dysfunctions, are unfortunately unable to slow down neurodegeneration [15,16].

In recent decades, it has been shown that AChE itself promotes the formation of Aβ fibrils and plaques [17], and this property of AChE results from interaction between Aβ and the peripheral anionic site (PAS) of the enzyme [18]. Dual binding site inhibitors, i.e., bifunctional inhibitors that simultaneously interact with both the catalytic site and PAS of AChE, preclude AChE-induced Aβ aggregation and may act as disease-modifying agents with multiple functions, simultaneously improving cognition and slowing down the rate of Aβ-induced neural degeneration [19,20,21,22]. The use of dual binding site AChE inhibitors in the treatment of transgenic mice resulted in an improvement in cognition along with a reduction in the level of brain amyloid plaques [23,24]. Nevertheless, the assortment of AChE PAS ligands, as potential drugs that have been validated in animal models of AD with amyloidosis, is still extremely limited.

Recently, we developed a new class of selective mammalian dual binding site AChE inhibitors with an acyclic and macrocyclic structure based on 1,3-bis[ω-(substituted benzylethylamino) alkyl]-6-methyluracils. In particular, novel 6-methyl uracil derivatives with ω-(substituted benzylethylamino) alkyl chains at the N atoms of the pyrimidine ring (Figure 1) were reported as selective AChE inhibitors [25,26]. In these compounds, the substituted electron-withdrawing nitro-, trifluoro-, and fluoro-group benzylethylamino moieties were bridged to N atoms of the 6-methyluracil moiety via various polymethylene chains. The most active of these compounds, **1a**–**f**, with nitro and trifluoro substituents, are depicted in Figure 1. Molecular modeling and X-ray structures demonstrated that the compounds **1a**–**f** bound AChE as bifunctional inhibitors, blocking the entrance of the gorge of the enzyme with the substituted benzylethylamino group (blue in Figure 1) and masking the PAS area with the 1,3-bis(alkyl)-6-methyluracil moiety (red in Figure 1). This mode of action of the compounds **1a**–**f** resulted in remarkable selectivity for human AChE (hAChE). Some of the compounds demonstrated inhibitory power in the nanomolar range, 10,000 or more times higher than for human butyrylcholinesterase (hBChE). In vivo experiments showed that the most potent AChE inhibitor **1b** improved working memory and significantly reduced the number and area of Aβ plaques in the brain of double transgenic mice expressing a chimeric mouse/human amyloid precursor protein and a mutant human presenilin 1 (APP/PS1) [25,26].

In this report, we expanded this series of AChE inhibitors based on 1,3-bis[ω-(substituted benzylethylamino)alkyl] uracils in the search for effective AChE inhibitors capable of improving cognitive dysfunctions caused by AD. In particular, we were interested in how the substitution with electron-withdrawing nitrile groups in compounds with a similar structure to compounds **1a**–**f** would potentially influence their activity against AChE. Nitrile groups have been observed to have a particular effect on biological activity [27], and in this study, we isolated a separate series of uracil-based compounds **2a**–**c** (Figure 1) with the same structure as compounds **1a**–**f** with the exception of cyano-substitutions of benzyl moieties.

In order to elucidate structure–activity relationships, the structure of the series of synthesized compounds was varied in several ways. For compounds **2a**–**c**, the polymethylene chains were varied, including tetra-, penta-, and hexamethylene chains. The choice of these chain lengths was based on our previous studies of compounds of similar structure, in particular, compounds **1a**–**f** and the X-ray structure of **1b**/hAChE complex [25,26], which indicated these to be optimal. In addition, the tertiary amino moiety and 6-methyluracil ring were replaced by a secondary amino moiety and a condensed uracil derivative, respectively, to obtain quinazoline-2,4-dione, as depicted for compounds **3** and **4a**,**b** (Figure 1). The secondary NH amino group and the expanded aromatic system of condensed uracil, quinazoline-2,4-dione, can result in increased binding affinity to the enzyme due to H-donor hydrogen bonding and π–π interactions with amino acid residues. Further, tertiary amino groups of some of the compounds synthesized, namely, compounds **2b**,**c**, **4b,** were converted into ammonium moieties by hydrohalogenation to afford dihydrobromides **2b·**2HBr, **2c·**2HBr and **4b·**2HBr.

Herein, we present the synthesis and biological evaluation in vitro and in vivo of a series of new AChE inhibitors based on 1,3-bis [ω-(substituted benzylethylamino)alkyl] uracil derivatives.

## 2. Results and Discussion

### 2.1. Preparation of 1,3-Bis[ω-(Substituted Benzylethylamino)Alkyl] Uracils

The synthetic strategy for targeting uracils **2a**–**c** with varying polymethylene chain lengths is depicted in Scheme 1. This began with the amination of dibromides **7a**–**c** and was followed by the alkylation of bisamines **8a**–**c** with *o*-nitrilebenzyl bromide. This strategy was developed for the synthesis of compound **1 [25]**, and the procedure for the synthesis of initial reagents **7a**–**c**, **8a**–**c** is described elsewhere [28,29].

The synthetic protocol can be expanded to condensed uracil derivatives, especially quinazoline-2,4-dione. As shown above, the first step in the synthesis of quinazoline-2,4-diones **4a**,**b** is substitution of terminal atoms of Br in dibromides **9a**,**b** with ethylamino groups to obtain bisamines **10a**,**b**. Reaction of bisamines **10a**,**b** with *o*-nitrilebenzyl bromide gave the intended compounds **4a**,**b** (Scheme 2).

Contrarily, for the synthesis of compound **3** with secondary amino group in pentamethylene chains a different approach was used. It was evident that compound **3** had to be prepared starting from uracil derivative with terminal primary amino groups in chains at pyrimidine N atoms. Attempt of direct substitution of Br in dibromide **5b** by NH_2_ in ammonia solution was unsuccessful. Then dibromide **5b** was converted to azide **11 [30]** and the Staudinger reaction [31] of the azide with triphenylphosphine gave bisamine **12**. The aimed secondary bisamine **3** was prepared reacting **12** with 2-nitrilebenzaldehyde and reducing intermediate aldoimine **13** with NaBH_4_ (Scheme 3). It is worth noting that direct alkylation of bisamine **12** with *o*-nitrilebenzyl bromide produced not compound **3** but compound with fully alkylated amino-groups.

Uracils **2a**–**c**, **3**, and quinazoline-2,4-diones **4a,b** are insoluble in water, and hydrohalogenation of the N atoms in chains resulted in water-soluble forms of these compounds. Specifically, the hydrobromination of compounds **2b**,**c** and **4b** in alcohol produced dihydrobromides **2b·**2HBr, **2c·**2HBr, and **4b·**2HBr, respectively.

### 2.2. The Inhibitory Effects on Cholinesterases and Acute Toxicity of Uracils and Quinazoline-2,4-Diones

The inhibitory activities of compounds **2a**–**c**, **3**, **4a**,**b**, **5a**,**b**, **6**, and **13** against hAChE and hBChE were evaluated according to Ellman’s method [32], and the results for the the 50% inhibitory concentration (IC_50_) values of all compounds and their selectivity indexes for AChE over BChE are summarized in Table 1. The acute toxicity of the compounds, discussed in terms of lethal doses (LD_50_) for mice, are also represented in the table.

For comparison with the discussed compounds, the table also shows the data for inhibitors **1a**–**c** and **1d**–**f**, as previously published [25]. Target compounds **2a**–**c** with the nitrile group and different lengths of polymethylene chains exhibited almost equal inhibitory activity against hAChE, which decreased with the increase in the number of methylene groups from 4 to 6 less dramatically than was the case for the activities of compounds **1a**–**c** and **1d**–**f**. In these series of compounds, **1a**–**c**, **1d**–**f**, **2a**–**c**, the inhibitory activity against hAChE at elongation of chains from 4 to 5 methylene groups first increased and with further elongation of chains from 5 to 6 methylene groups decreased. It needs to be noted that nitriles **2a**–**c** were the most active against hAChE in the series of the compounds (6-methyluracil derivatives), although selectivity of some of the compounds with nitro- and trifluoromethyl groups for hAChE vs. hBChE was higher than that of nitriles **2a**–**c**. In general, this selectivity of the inhibitors **1a**–**f** and **2a**–**c** was significantly higher than that of donepezil hydrochloride. The acute toxicity of nitriles **2a**–**c** was significantly less than the toxicity of nitro- and trifluoromethylcompounds and the reference drug.

A different situation was observed for nitriles **4a**,**b** with condensed uracil moieties. When going from the 6-methyluracil to the quinazoline-2,4-dione moiety with the same chain length (compounds **2a** and **4a**), the activity toward hAChE sharply decreased by three orders of magnitude and there was less selectivity for the enzyme in comparison with BChE. An increase in polymethylene *N*_pyrimidine_-*N*-chain lengths by up to 5 methylene groups in compound **4b** markedly increased the strength of hAChE inhibition and its selectivity, in addition to a decrease in LD_50_. The most optimal chain length in the discussed series of compounds **1a**–**c**, **1d**–**f**, **2a**–**c**, **4a,b**, and **2b·**2HBr and **2c·**2HBr is a length of five methylene groups. Compounds with pentamethylene chains **1b**,**e**, **2b**, **4b**, and **2b·**2HBr exhibited the highest affinity towards hAChE. Molecular docking supported these results, particularly for quinazoline-2,4-diones **4a**,**b**. Quaternization of atoms of N in nitriles **2b**,**c** with HBr acid (compounds **2b·**2HBr and **2c·**2HBr) resulted in a slight increase of inhibitory power against hAChE, a decrease in selectivity, and a significant decrease in LD_50_. Contrarily, quinazoline-2,4-dione **4b·**2HBr with quaternized N atoms in pentamethylene chains compared to neutral counterpart **4b** demonstrated almost the same inhibitory activity against enzymes studied but significantly lower acute toxicity. If lower LD_50_ values for charged nitriles with 6-methyluracil moieties **2b·**2HBr and **2c·**2HBr compared to LD_50_ of neutral nitriles **2b,c** can be explained by a higher solubility of charged nitriles in water, then a significant increase in LD_50_ of charged nitrile with quinazoline-2,4-dione moiety **4b·**2HBr compared to neutral nitrile **4b** cannot be due to the difference in solubility in water. Studies of the pharmacokinetics of these compounds are necessary to highlight the discrepancy in acute toxicity of charged and neutral nitriles.

### 2.3. Molecular Docking Study of Title Compounds

Influence of quaternization of the aliphatic N atoms, replacement of 6-methyluracil moiety with quinazoline-2,4-dione and polymethylene chains’ length of nitriles obtained on interactions with hAChE gorge, was assessed by molecular docking. Positions with the best binding affinities (see Appendix A are discussed below. The position of the nitrile **2b** binding in the active site gorge of hAChE obtained by molecular docking is very close to that of its nitro group-containing analog **1b**, as observed by X-ray crystallography [26]. The inhibitor molecule spans through the whole gorge and occupies both the PAS and active site (Figure 2A). The 20 Å-long hAChE gorge leading from the enzyme surface to the catalytic triad (S203–H447–E334) is divided in the middle with a so-called bottleneck (formed by Y341, Y337, and Y124) into the peripheral anionic site and the active site [33]; the inhibitor occupies both of these sites.

The position of 6-methyluracil **2b·**2HBr with quaternized N atoms is similar but, additionally, has other specific interactions within the gorge, while for 6-methyluracil **2b**, hydrogen bonding is observed between the cyano group and the Y124 side chain in addition to π–π stacking between the 6-methyluracil ring and the Y341 side chain (Figure 2A). For 6-methyluracil **2b·**2HBr, π–cation interactions are observed between the quaternized N atom and Y337 and F338 and π–π stacking between the nitrile-benzyl ring and the W286 side chain (Figure 2A). E292 could also form a salt bridge with the second quaternized N atom, though this could not be observed with rigid protein molecular docking but was instead observed in molecular dynamics’ simulations performed for compound **1b** with a similar structure [26]. Due to this, quaternization of nitrogen atoms did not have a pronounced effect on binding strength.

Due to the high flexibility of the ligands, the conformation of the ligands inside the wider hBChE active site gorge is rather different from the one inside hAChE (Figure 2A,B). While the compounds are in elongated conformation and span through the full length of the narrow hAChE gorge, they are curled at the lower part of the hBChE gorge, and the difference in positions of noncharged and charged compounds is more pronounced (Figure 2B). The only specific interaction observed for 6-methyluracil **2b** was hydrogen bonding between the nitrile group and the enzyme oxyanion hole, while for 6-methyluracil, the **2b·**2HBr nitrile group in the oxyanion hole was replaced by a 6-methyluracil ring and, additionally, there were salt bridges between the charged amino group and D70 and hydrogen bonding between the second charged amino group and the P285 side chain.

The positions of quinazoline-2,4-dione derivatives **4b** and **4b·**2HBr inside hAChE did not differ much, with almost nearly identical specific interactions observed (Figure 3A). Binding to hBChE, on the contrary, was more greatly affected by replacement of the 6-methyluracil fragment with quinazoline-2,4-dione (Figure 3B); binding to the oxyanion hole was no longer observed. No specific interactions were seen for quinazoline-2,4-dione **4b**, and binding was achieved due to geometrical fitness, while in its charged counterpart, quinazoline-2,4-dione **4b·**2HBr, interactions were observed between one quaternized N atom and D70 and Y332, and the second quaternized N atom formed π–cation stacking interactions with the W82 side chain.

For quinazoline-2,4-dione derivatives, linker length affected binding affinity to hAChE. Molecular docking revealed that compound **4a** had fewer specific interactions than compound **4b** (Figure 4). The comparison of docked poses obtained with AutoDock 4.2.6 and AutoDock Vina 1.1.2 is presented in the Appendix A.

### 2.4. Behavioral Tests

From the structure–activity profile of the compounds discussed, nitriles **2a**–**c** with the 6-methyluracil moiety and nitrile **4b·**2HBr with the quinazoline-2,4-dione moiety seem to be more promising for inhibition of AChE in vivo.

LD_50_ values of the most active nitriles **2a**,**b** were used to indicate the ability of the compounds to cross the blood–brain barrier (BBB) and inhibit brain AChE after intraperitoneal injection of the LD_50_ dose. Therefore, the whole brains were removed 30 min after injection of the tested compound. The brain homogenates were prepared and AChE activity was measured spectrophotometrically according to the Ellman method [32]. The most effective in vitro AChE inhibitors **2a**,**b** exhibited significant in vivo potency for inhibiting brain AChE with 92 ± 3%. In turn, nitro- and trifluoromethyl compounds **1a**–**f** inhibited brain AChE with a maximum of 71 ± 1% activity [25].

Thus, the high activity of the compounds with uracil and quinazoline moieties against AChE was established by in vitro experiments measuring their inhibitory effect, and molecular modeling showed that these compounds bind to the enzyme as dual binding site inhibitors in the PAS and active center. Among the active compounds, the most effective inhibitor nitrile **2b** was selected for in vivo study, namely for treatment of memory impairment on the AD model. In particular, the effect of compound **2b** on spatial memory was evaluated in the scopolamine mouse model of AD.

The reward alternation task using T-maze was performed to evaluate whether compound **2b** could attenuate the cognitive deficits in scopolamine-treated mice. Mice were treated with scopolamine and tested compounds 40 and 20 min before T-maze test, respectively. The percentage of correct choices, percentage of mice reaching the criterion of learning, and dynamic of reaching the criterion of learning were evaluated. Scopolamine-injected mice showed significant impairment of spatial memory in a T-maze, which was characterized by decreases in percentages of both correct choice and reaching the learning criterion at day 14. This memory deficit was rescued, to some extent, by treatment with either compound **2b** (1 and 5 mg/kg) or donepezil (1 mg/kg). The most effective rescue of memory impairment, as shown by restoration of correct choice and reaching the learning criterion at day 14, was found in the case of treatment with compound **2b** at a dose of 5 mg/kg, with other doses being less effective. Notably, a higher dose of compound **2b** (10 mg/kg) produced less therapeutic effects in terms of spatial memory deficit. Apparently, this decrease in the effectiveness of the higher dose of compound **2b** could be associated with cholinergic side effects, such as gastrointestinal upset, interfering with task learning during behavioral tests [34]. However, 5 mg/kg nitrile **2b** showed the same efficacy as donepezil (1 mg/kg) in the rescue of spatial memory deficit in scopolamine-injected mice (*p* > 0.05) (Figure 5A–C). Scopolamine-treated mice had decline in dynamic of reaching the criterion of learning. In contrast, mice treated with nitrile **2b** (1 and 5 mg/kg), similar to donepezil-treated mice, showed a progressively increased trend of learning, comparable to the control group (Figure 5C).

## 3. Experimental Section

### 3.1. Chemistry

#### 3.1.1. General Methods

The NMR experiments were carried out using Bruker spectrometer AVANCE-400 (400.1 MHz (^1^H), 100.6 MHz (^13^C), Bruker BioSpin, Germany). Matrix-Assisted Laser Desorption/Ionization Time of Flight mass spectra (MALDI-MS) were recorded on a Bruker ULTRAFLEX III mass spectrometer (Bruker Daltonic GmbH, Germany) using *p*-nitroaniline as a matrix. The IR spectra of compounds were recorded on a Vector 22 FTIR Spectrometer (Bruker, Germany) in the 4000 to 400 cm^−1^ range at a resolution of 1 cm^−1^. Microelemental analysis data were obtained on a CHN-O analyzer (Eurovector, S.p.A., Italy) and were within ±0.3% of theoretical values for C, H, and N. Thin layer chromatography was performed on Silufol-254 plates (Sigma-Aldrich, Germany, the solvent system was diethyl ether or ethylacetate/methanol mixture); visualization of spots was carried out under UV light (λ = 254 nm). For column chromatography, silica gel of 60 mesh from Fluka (Fluka, Germany) was used. All solvents were dried according to standard protocols.

#### 3.1.2. Initial Compounds for Preparation of Title Cholinesterase Inhibitors

Synthesis of initial compounds 1,3-bis(4-bromobutyl)-6-methyluracil (**7a**), 1,3-bis(5-bromopentyl)-6-methyluracil (**7b**), 1,3-bis(6-bromohexyl)-6-methyluracil (**7c**) [35], 1,3-bis(4-ethylaminobutyl)-6-methyluracil (**8a**), 1,3-bis(5-ethylaminopentyl)-6-methyluracil (**8b**), 1,3-bis(6-ethylaminohexyl)-6-methyluracil (**8c**) [28,29], 1,3-bis(5-azidopentyl)-6-methyluracil (**11**) [30], and 1,3-bis(5-bromopentyl)quinazoline-2,4-dione (**9b**) was carried out as previously reported [36].

*1,3-bis(4-Bromobutyl)-quinazoline-2,4-dione* (**9a**), NaH (2.50 g, 104.2 mmol) was added to a suspension of 1,2,3,4-tetrahydroquinazoline-2,4-dione (10.1 g, 62.6 mmol) in DMF (150 mL), and the mixture was stirred at 50–55 °C for 2 h. A solution of 1,4-dibromobutane (108.2 g, 500.8 mmol) in DMF (50 mL) was added under stirring, and the mixture was stirred for 55–65 °C until neutral reaction of an aqueous solution, with withdrawal of a sample from the mixture for monitoring (10 h). The solvent and excess 1,4-dibromobutane were removed under reduced pressure, the residue was treated with chloroform, the mixture was filtered, and the filtrate was concentrated to a volume of 15–20 mL and subjected to column chromatography on SiO_2_. The column was eluted in succession with petroleum ether and diethyl ether/petroleum ether (1:1). Elution with the solvent mixture 10:1 gave 9.50 g (35%) of dibromide **9a** as an oily substance. IR (neat, cm^−1^) maximum wavenumber ν_max_: 2960, 1702, 1658, 1609, 1484, 1403, 1036, 759; ^1^H NMR (CDCl_3_) δ 8.17–8.15 (multiplet m, 1H, benzene ring protons ArH), 7.65–7.61 (m, 1H, ArH), 7.22–7.16 (m, 2H, 2ArH), 4.14–4.10 (t, 2H, N^3^_uracil_CH_2_, *J* 7.2 Hz), 4.08–4.04 (triplet t, 2H, N^1^_uracil_CH_2_, *J* 7.1 Hz), 3.45-3.38 (m, 4H, 2BrCH_2_), 1.97–1.80 (m, 8H, 2CH_2_). The ^13^C NMR δ: 160.61, 149.80, 138.64, 134.24, 128.24, 121.98, 114.74, 112.57, 41.75, 39.88, 32.27, 32.08, 29.21, 28.78, 25.68, 25.03. MALDI-MS (mass-to-charge ratio *m/z*): calculated (calcd) for C_16_H_20_Br_2_N_2_O_2_ [M + H]^+^ 433.0, found: 433.0. Analytically calculated (Anal. Calcd) for C_16_H_29_Br_2_N_2_O_2_: C, 44.47; H, 4.66; Br, 36.98; N, 6.48. Found: C, 44.51; H, 4.62; Br, 37.06; N, 6.38.

Synthesis of bisamines **10a**,**b** (Scheme 2). General procedure. Dibromide **9a**,**b** (12.0 mmol) was added to a 20% EtNH_2_ solution in 2-propanol (100 mL). The reaction mixture was kept at room temperature (RT) for 2 days and then concentrated in vacuo. A solution of MeONa, which was prepared from Na (0.55 g, 24.0 mmol) in MeOH (30 mL), was added to the residue. The solvent was evaporated in vacuo, and the reaction product was extracted with diethyl ether (2 × 50 mL). The ether was removed under reduced pressure to obtain the intended compound.

*The 1,3-bis(4-Ethylaminobutyl)-quinazoline-2,4-dione* (**10a**), Yield 86%; oil; IR (neat, cm^−1^) ν_max_: 3388, 2930, 2863, 1700, 1658, 1605, 1482, 1405, 1110, 758; ^1^H NMR δ 8.18-8.16 (m, 1H, ArH), 7.62–7.58 (m, 1H, ArH), 7.20–7.16 (m, 2H, 2ArH), 4.11–4.04 (m, 4H, N^3^_uracil_CH_2_, N^1^_uracil_CH_2_), 2.65–2.59 (m, 10H, 4NCH2, 2NH)_,_ 1.74–1.67 (m, 4H, 2CH_2_), 1.60–1.52 (m, 4H, 2CH_2_), 1.08–1.03 (m, 6H, 2CH_3_). The ^13^C NMR (CDCl_3_) δ: 161.10, 150.11, 139.27, 134.47, 128.41, 122.00, 115.07, 112.70, 48.76, 48.14, 43.26, 43.57, 43.11, 41.28, 28.97, 28.89, 27.18, 26.71, 24.24, 24.01, 14.42, 14.38. MALDI-MS (*m/z*): calcd for C_20_H_32_N_4_O_2_ [M + H]^+^ 361.3, found: 361.5. Anal. Calcd for C_22_H_36_N_4_O_2_: C, 66.63; H, 8.95; N, 15.54. Found: C, 66.59; H, 8.91; N, 15.51.

*The 1,3-bis(5-Ethylaminopentyl)-quinazoline-2,4-dione* (**10b**), Yield 90%; oil; IR (neat, cm^−1^) ν_max_: 3391, 2932, 2860, 1702, 1659, 1609, 1485, 1404, 1116, 760; ^1^H NMR (CDCl_3_) δ 8.22-8.20 (m, 1H, ArH), 7.65–7.61 (m, 1H, ArH), 7.26–7.14 (m, 2H, 2ArH), 4.11–4.04 (m, 4H, N^3^_uracil_CH_2_, N^1^_uracil_CH_2_), 2.64–2.60 (m, 10H, 4NCH2, 2NH)_,_ 1.74–1.69 (m, 4H, 2CH_2_), 1.56–1.52 (m, 4H, 2CH_3_), 1.48–1.40 (m, 4H, 2CH_3_), 1.13–1.06 (m, 6H, 2CH_3_). The ^13^C NMR δ: 161.15, 150.21, 139.22, 134.43, 128.58, 122.16, 115.23, 112.99, 48.86, 48.24, 43.46, 43.37, 43.08, 41.23, 28.97, 28.89, 27.18, 26.71, 24.24, 24.01, 14.42, 14.38. MALDI-MS (*m/z*): calcd for C_22_H_36_N_4_O_2_ [M + H]^+^ 389.3, found: 389.2. Anal. Calcd for C_22_H_36_N_4_O_2_: C, 68.01; H, 9.34; N, 14.42. Found: C, 67.94; H, 9.41; N, 14.45.

#### 3.1.3. Synthesis of Cholinesterase Inhibitors, 1,3-Bis[ω-(*O*-Nitrilebenzylethylamino)Alkyl]-6-Methyluracils and Quinazoline-2,4-Diones

Synthesis of compounds **2a**–**c**, **4a**,**b** (Scheme 1 and Scheme 2). General procedure: A mixture of bisamine **8a**–**c, 10a,b** (5.0 mmol), *ortho*-cyanobenzyl bromide (10.0 mmol) and potassium carbonate (3.45 g, 25.0 mmol) was stirred in CH_3_CN (150 mL) at 60–65 °C for 12 h. The precipitate was filtered out. The solution was concentrated to 10–15 mL and transferred to a column with SiO_2_. The column was successively washed with petroleum ether and 60:1 and 40:1 chloroform/methanol mixtures. The target compounds **2a**–**c**, **4a**,**b** were isolated from the 40:1 chloroform/methanol mixture fractions.

*The 1,3-bis[4-(o-Nitrilebenzylethylamino)butyl]-6-methyluracil* (**2a**), Yield 52%; oil; IR (neat, cm^−1^) ν_max_: 3067, 2967, 2808, 2223, 1700, 1661, 1449, 1363, 1210, 1064, 767; ^1^H NMR (CDCl_3_) δ 7.63–7.60 (m, 3H, 3ArH), 7.55–7.52 (m, 3H, 3ArH), 7.33–7.31 (m, 2H, 2ArH), 5.54 (singlet s, 1H, C^5^_uracil_H), 3.90–3.88 (t, 2H, N^3^_uracil_CH_2_, *J* 7.1 Hz), 3.78–3.74 (m, 6H, N^1^_uracil_CH_2_, 2CH_2_Ph), 2.55–2.49 (m, 8H, 4NCH_2_)_,_ 2.20 (s, C^6^_uracilC_CH_3_), 1.64–1.60 (m, 4H, 2CH_2_), 1.54–1.50 (m, 4H, 2CH_2_), 1.06–1.02 (m, 6H, 2CH_3_). The ^13^C NMR δ: 162.11, 151.75, 150.87, 132.41, 129.71, 127.30, 117.59, 117.59, 112.15, 111.57, 101.38, 55.60, 52.45, 46.98, 44.98, 40.66, 26.33, 25.22, 24.04, 19.74, 11.12. MALDI-MS (*m/z*): calcd for C_33_H_42_N_6_O_2_ [M − H]^+^ 553.3, found: 553.0. Anal. Calcd for C_33_H_42_N_6_O_2_: C, 71.45; H, 7.63; N, 15.15. Found: C, 71.52; H, 7.58; N, 15.08.

*The 1,3-bis[5-(o-Nitrilebenzylethylamino)pentyl]-6-methyluracil* (**2b**), Yield 61%; oil; IR (neat, cm^−1^) ν_max_: 3068, 2967, 2936, 2861, 2807, 2223, 1701, 1662, 1449, 1361, 1209, 1054, 766; ^1^H NMR (CDCl_3_) δ 7.64–7.52 (m, 8H, 8ArH), 5.54 (s, 1H, C^5^_uracil_H), 3.89–3.86 (t, 2H, N^3^_uracil_CH_2_, *J* 7.5 Hz), 3.78–3.74 (m, 6H, N^1^_uracil_CH_2_, 2CH_2_Ph), 2.56–2.52 (m, 4H, 2NCH_2_)_,_ 2.49–2.44 (m, 4H, 2NCH_2_), 2.21 (s, C^6^_uracilC_CH_3_), 1.61–1.57 (m, 4H, 2CH_2_), 1.53–1.48 (m, 4H, 2CH_2_), 1.35–1.32 (m, 4H, 2CH_2_), 1.06–1.02 (m, 6H, 2CH_3_).The ^13^C NMR δ: 161.45, 151.82, 151.92, 144.16, 132.49, 129.97, 126.59, 117.75, 112.40, 112.11, 101.05, 52.60, 47.13, 44.38, 40.09, 27.93, 27.38, 26.29, 24.23, 23.94, 19.43, 10.37. MALDI-MS (*m/z*): calcd for C_35_H_46_N_6_O_2_ [M − H]^+^, [M − C_2_H_5_]^+^, [M − CH_2_PhCN]^+^ 581.4, 553.3, 466.3, found: 581.3, 553.2, 466.1. Anal. Calcd for C35H46N6O2: C, 72.13; H, 7.96; N, 14.42. Found: C, 72.00; H, 8.02; N, 14.34.

*The 1,3-bis[6-(o-Nitrilebenzylethylamino)hexyl]-6-methyluracil* (**2c**), Yield 56%; oil; IR (neat, cm^−1^) ν_max_: 3067, 2966, 2934, 2858, 2807, 2224, 1700, 1661, 1449, 1363, 1210, 1059, 766; ^1^H NMR (CDCl_3_) δ 7.63–7.52 (m, 6H, 8ArH), 7.36–7.28 (m, 2H, 2ArH), 5.54 (s, 1H, C^5^_uracil_H), 3.89-3.85 (t, 2H, N^3^_uracil_CH_2_, *J* 7.5 Hz), 3.78–3.74 (m, 6H, N^1^_uracil_CH_2_, 2CH_2_Ph), 2.56–2.52 (m, 4H, 2NCH_2_)_,_ 2.47–2.44 (m, 4H, 2NCH_2_), 2.21 (s, C^6^_uracilC_CH_3_), 1.60–1.57 (m, 4H, 2CH_2_), 1.47–1.44 (m, 4H, 2CH_2_), 1.33–1.29 (m, 8H, 4CH_2_), 1.06–1.02 (m, 6H, 2CH_3_). The ^13^C NMR δ: 161.99, 152.07, 150.60, 144.52, 132.77, 132.54, 129.79, 127.24, 118.01, 112.77, 101.56, 56.06, 53.19, 47.44, 45.27, 44.78, 40.63, 28.89, 27.25, 26.04, 19.08, 11.59. MALDI-MS (*m/z*): calcd for C_37_H_50_N_6_O_2_ [M − H]^+^ 609.4, found: 609.4. Anal. Calcd for C_37_H_50_N_6_O_2_: C, 72.75; H, 8.25; N, 13.76. Found: C, 72.68; H, 8.28; N, 13.71.

*The 1,3-bis[4-(o-Nitrilebenzylethylamino)butyl]-quinazoline-2,4-dione* (**4a**), Yield 65%; oil; IR (neat, cm^−1^) ν_max_: 2966, 2937, 2868, 2810, 2224, 1767, 1702, 1657, 1609, 1485, 1403, 1355, 1288, 1045, 761; ^1^H NMR (CDCl_3_) δ 8.07 (dublet d, 1H, ArH, *J* 1.5 Hz), 7.60-7.45 (m, 7H, 7ArH), 7.26–7.12 (m, 4H, 4ArH), 3.96–3.93 (m, 4H, 2N_quinazoline_CH_2_), 3.63 (s, 4H, 2CH_2_Ph), 2.44–2.40 (m, 8H, 4NCH_2_), 1.64–1.60 (m, 2H, 2CH_2_), 1.48–1.44 (m, 2H, 2CH_2_), 0.94–0.91 (m, 6H, 2CH_3_). The ^13^C NMR δ: 161.09, 150.167, 143.97, 143.71, 139.11, 134.46, 132.33, 131.87, 128.97, 128.06, 126.35, 122.60, 121.43, 117.09, 115.36, 113.30, 112.12, 55.46, 52.35, 47.36, 43.66, 41.17, 25.24, 24.11, 23.34, 11.18. MALDI-MS (*m/z*): calcd for C_36_H_42_N_6_O_2_ [M + H]^+^ 591.3, found: 591.4. Anal. Calcd for C_36_H_42_N_6_O_2_: C, 73.19; H, 7.17; N, 14.23. Found: C, 73.00; H, 7.25; N, 14.28.

*The 1,3-bis[5-(o-Nitrilebenzylethylamino)pentyl]-quinazoline-2,4-dione* (**4b**), Yield 59%; oil; IR (neat, cm^−1^) ν_max_: 2966, 2937, 2868, 2810, 2224, 1767, 1702, 1657, 1609, 1485, 1403, 1355, 1288, 1045, 761; ^1^H NMR (CDCl_3_) δ 8.11 (d, 1H, ArH, *J* 1.2 Hz), 7.56–7.44 (m, 7H, 7ArH), 7.26–7.12 (m, 4H, 4ArH), 4.02–3.97 (m, 4H, 2NquinazolineCH_2_), 3.66 (s, 4H, 2CH_2_Ph), 2.48–2.40 (m, 8H, 4NCH_2_), 1.63–1.60 (m, 4H, 2CH_2_), 1.47–1.44 (m, 4H, 2CH_2_), 1.38–1.30 (m, 4H, 2CH_2_), 0.98–0.95 (m, 6H, 2CH_3_).The ^13^C NMR δ: 161.80, 151.12, 144.43, 139.86, 135.13, 132.80, 129.74, 128.98, 127.24, 122.58, 117.81, 115.70, 113.60, 112.30, 56.19, 52.54, 48.04, 43.61, 41.73, 27.68, 27.06, 26.73, 24.92, 11.67. MALDI-MS (*m/z*): calcd for C_38_H_46_N_6_O_2_ [M − H]^+^ 617.4, [M + Na]+ 641.4, found: 617.5, 641.5. Anal. Calcd for C_38_H_46_N_6_O_2_: C, 73.76; H, 7.49; N, 13.58. Found: C, 73.85; H, 7.43; N, 13.56.

*The 1,3-bis(5-Aminopentyl)-6-methyluracil dihydrochloride* (**12**) (Scheme 3), A mixture of bisazide **11** (1.8 g, 2.9 mmol) and triphenylphosphine was stirred in THF (60 mL) at RT for 16 h. The solvent was removed under reduced pressure. A mixture of ethyl acetate (50 mL) and 15% aqueous solution of HCl (30 mL) was added to the residue. The resulting aqueous layer was separated and washed with ethyl acetate (3 × 30 mL). The solvent was evaporated. The residue was thoroughly triturated in anhydrous diethyl ether, the ether was decanted, and the residue was dried to obtain 0.88 g (83%) bisamine **12** as white powder. Melting point (M.p.) 200 °C; IR (KBr pellet, cm^−1^) ν_max_: 3431, 3058, 2965, 2873, 1698, 1645, 1472, 1146, 1038, 849; ^1^H NMR (DMSO-d_6_) δ 8.10–8.06 (m, 6H, 2NH_2_ × 2HCl), 5.59 (s, 1H, C^5^_uracil_H), 3.75 (m, 4H, N^3^_uracil_CH_2_, N^1^_uracil_CH_2_), 2.72 (m, 4H, 2NH_2_CH_2_), 2.25 (s, 3H, C^6^_uracil_CH_3_),1.59 (m, 8H, 4CH_2_), 1.28 (m, 4H, 2CH_2_). The ^13^C NMR δ: 161.16, 152.83, 151.13, 99.80, 44.47, 27.85, 26.64, 23.52, 22.66, 18.68. Anal. Calcd for C_15_H_30_Cl_2_N_4_O_2_: C, Anal. Calcd for C_15_H_30_Cl_2_N_4_O_2_: C, 50.99; H, 8.56; Cl 20.07; N, 15.86. Found: C, 50.90; H, 8.54; Cl 20.13; N, 15.96.

*The 1,3-bis[5-(o-Nitrilebenzyl)aldiminopentyl]-6-methyluracil* (**13**), Bisamine **12** (0.88 g, 2.4 mmol) was dissolved in a solution of sodium (0.11 g, 4.8 mmol) in methanol (30 mL), and the resulting mixture was stirred at RT for 30 min. Methanol was removed under reduced pressure, the residue was treated with chloroform (30 mL), and the precipitate formed was filtered. Next, 2-cyanobenzyl bromide (0.62 g, 4.8 mmol) was added and the reaction mixture was stirred at RT for 6 h. Evaporation of the solvent gave 1.16 g (93%) of the compound **13** as cream powder. M.p. 95° C; IR (KBr pellet, cm^−1^) ν_max_: 2962, 2858, 2220, 1695, 1658, 1469, 1261, 1202, 1095, 1020, 799; ^1^H NMR (DMSO-d_6_) δ: 7.61–7.45 (m, 8H, 8ArH), 5.53 (s, 1H, C^5^_uracil_H), 5.33 (br. s, 2H, 2CH), 4.17 (m, 4H, N^3^_uracil_CH_2_, N^1^_uracil_CH_2_), 2.45–2.35 (m, 4H, 2NCH_2_), 1.51–1.25 (m, 12H, 6CH_2_). The ^13^C NMR (DMSO-d6) δ: 162.73, 152.55, 151.69, 145.99, 133.41, 131.79, 129.40, 123.71, 121.71, 99.96, 69.94, 44.49, 43.07, 29.61, 27.90, 27.06, 24.71, 23.96, 23.35, 19.05. MALDI-MS (*m/z*): calcd for C_31_H_34_N_6_O_2_ [M − H]^+^ 521.3, found: 521.3. Anal. Calcd for C_31_H_34_N_6_O_2_: C, 71.24; H, 6.56; N, 16.08. Found: C, 71.12; H, 6.62; N, 16.14.

*The 1,3-bis[5-(o-Nitrilebenzylamino)pentyl]-6-methyluracil* (**3**), Sodium borohydride (0.31 g, 8.0 mmol) was added solution of bisaldimine **13** (1.16 g, 2.2 mmol) in methanol (30 mL) that had been cooled to 0 °C. The reaction mixture was stirred at RT for 1 h and at 50 °C for 2 h. Sodium hydroxide 0.1 N solution (20 mL) was added, followed by extraction and washing with chloroform. The organic layer was washed thrice with equal volumes (20 mL) of water, dried over sodium sulfate, concentrated, and transferred to a column with SiO_2_. The column was successively washed with ethyl acetate and 20:1 acetyl acetate/diethylamine mixture. Target compound **3** was isolated from the solvent mixture fractions as oil in a yield of 0.8 g (68%). IR (neat, cm^−1^) ν_max_: 3274, 2965, 2922, 2855, 2222, 1697, 1656, 1466, 1373, 1107, 749; ^1^H NMR (CDCl_3_) δ 7.80–7.77 (m, 2H, 2ArH), 7.60–7.54 (m, 4H, 4ArH), 7.48–7.45 (m, 2H, 2ArH), 5.53 (s, 1H, C^5^_uracil_H), 3.87–3.84 (m, 2H, N^3^_uracil_CH_2_), 3.77–3.74 (m, 2H, N^1^_uracil_CH_2_), 3.72–3.70 (m, 4H, 2CH_2_Ph), 2.55–2.49 (m, 6H, 2NHCH_2_, 2NH)_,_ 1.60-1.49 (m, 8H, 4CH_2_), 1.39–1.34 (m, 4H, 2CH_2_). The ^13^C NMR δ: 162.08, 153.13, 152.20, 146.72, 133.29, 132.03, 129.13, 124.19, 122.83, 100.72, 70.45, 70.29, 44.93, 44.12, 30.10, 29.85, 29.72, 28.46, 27.50, 24.60, 24.34, 19.50. MALDI-MS (*m/z*): calcd for C_31_H_38_N_6_O_2_ [M − H]^+^ 525.3, respectively, found: 525.4. Anal. Calcd for C_31_H_38_N_6_O_2_: C, 70.70; H, 7.27; N, 15.96. Found: C, 70.77; H, 7.14; N, 16.03.

Preparation of dihydrobromides **2b·**2HBr, **2c·**2HBr and **4b·**2HBr. General procedure: Concentrated hydrobromic acid (0.38 mL, 7.0 mmol) was added to a solution of title compound **2b**,**c**, **4b** (1.2 mmol) in methanol (40 mL). The reaction mixture was refluxed for 24 h. After cooling, the solvent was evaporated. The residue was triturated in anhydrous diethyl ether, the ether was decanted, and the residue was dried to afford the intended dihydrobromide.

*The 1,3-bis[5-(o-**Nitrilebenzylethylamino)pentyl]-6-methyluracil dihydrobromide* (**2b·**2HBr), Yield 78%; M.p. 77–79 °C; IR (KBr pellet, cm^−1^) ν_max_: 3402, 2943, 2625, 2227, 1696, 1654, 1469, 1361, 1289, 1218, 1049, 769; ^1^H NMR (CDCl_3_) δ: 11.10 (s, 2H, 2N^+^H), 8.52 (m, 2H, 2ArH), 7.85–7.54 (m, 6H, 6ArH), 5.56 (s, 1H, C^5^_uracil_H), 4.57 (br. s, 4H, 2CH_2_Ph), 3.97–3.76 (m, 4H, N^3^_uracil_CH_2_, N^1^_uracil_CH_2_), 3.30–3.10 (m, 8H, 4NCH_2_), 2.29 (s, C^6^_uracil_CH_3_), 2.06–1.98 (m, 4H, 2CH_2_), 1.75–1.19 (m, 14H, 4CH_2_, 2CH_3_). The 13C NMR δ: 161.67, 151.72, 150.66, 146.35, 133.81, 133.44, 132.29, 131.03, 128.63, 128.74, 125.11, 122.07, 116.45, 114.13, 101.62, 69.39, 54.31, 51.76, 47.61, 39.50, 28.55, 26.60, 23.74, 22.41, 20.29, 19.79. MALDI-MS (*m/z*): calcd for C_35_H_48_Br_2_N_6_O_2_ [M − HBr − H]^+^ 661.3, found: 661.2. Anal. Calcd for C_35_H_48_Br_2_N_6_O_2_: C, 56.46; H, 6.50; Br, 21.46; N, 11.29. Found: C, 56.50; H, 6.43; Br, 21.54; N, 11.35.

*The 1,3-bis[6-(o-Nitrilebenzylethylamino)hexyl]-6-methyluracil dihydrobromide* (**2c·**2HBr), Yield 90%; M.p. 54-56 °C; IR (neat, cm^−1^) ν_max_: 3424, 2938, 2862, 2623, 2227, 1694, 1654, 1470, 1432, 1364, 1217, 1050, 771; 1H NMR (CDCl3) δ: 11.71 (br. s, 2H, 2N^+^H), 8.71 (m, 2H, 2ArH), 7.84–7.75 (m, 4H, 4ArH), 7.63–7.61 (m, 2H, 2ArH), 5.60 (s, 1H, C^5^_uracil_H), 4.46 (br. s, 4H, 2CH_2_Ph), 3.93–3.90 (t, 2H, N^3^_uracil_CH_2_, *J* 7.5 Hz), 3.82–3.79 (t, 2H, N^1^_uracil_CH_2_, *J* 7.4 Hz), 3.23–3.05 (m, 8H, 4NCH2), 2.25 (s, C^6^_uracil_CCH3), 1.93–1.88 (m, 4H, 2CH_2_), 1.67–1.34 (m, 18H, 6CH_2_, 2CH_3_). The ^13^C NMR δ: 161.61, 151.50, 150.68, 133.74, 133.19, 131.83, 130.21, 116.51, 113.52, 100.39, 69.21, 53.33, 50.63, 46.51, 43.79, 39.66, 27.39, 26.57, 25.76, 25.22, 21.91, 19.18. MALDI-MS (*m/z*): calcd for C_37_H_52_Br_2_N_6_O_2_ [M − HBr − H]^+^ 689.3, [M − 2HBr − H]^+^ 609.4, found: 689.2, 609.2. Anal. Calcd for C_37_H_52_Br_2_N_6_O_2_: C, 57.52; H, 6.78; Br, 20.68; N, 10.88. Found: C, 57.57; H, 6.83; Br, 20.60; N, 11.00.

*The 1,3-bis[5-(o-Nitrilebenzylethylamino)pentyl]-quinazoline-2,4-dione dihydrobromide* (**4b·**2HBr), Yield 82%; M.p. 59–61 °C; IR (KBr pellet, cm^−1^) ν_max_: 3389, 2946, 2629, 2227, 1698, 1652, 1609, 1485, 1469, 1354, 1042, 761; ^1^H NMR (CDCl_3_) δ: 11.61 (s, 2H, 2N^+^H), 8.75 (m, 2H, 2ArH), 8.21 (m, 1H, ArH), 7.81–7.61 (m, 9H, 9ArH), 4.48 (br. s, 4H, 2CH_2_Ph), 4.24–4.08 (m, 4H, 2NquinazolineCH_2_), 3.23–3.10 (m, 8H, 4NCH_2_), 2.05 (m, 4H, 2CH_2_), 1.84–1.50 (m, 8H, 4CH_2_), 1.32–1.22 (m, 6H, 2CH_3_). The ^13^C NMR δ: 161.20, 150.41, 142.52, 139.29, 135.32, 134.87, 133.76, 133.53, 131.82, 131.51, 131.31, 129.00, 123.10, 116.65, 116.37, 115.55, 113.96, 95.70, 41.14, 27.61, 25.00, 23.31, 14.73, 11.05, 10.43. MALDI-MS (*m/z*): calcd for C_38_H_48_Br_2_N_6_O_2_ [M − 2HBr − H]^+^ 617.4, found: 617.4. Anal. Calcd for C_38_H_48_Br_2_N_6_O_2_: C, 58.47; H, 6.20; Br, 20.47; N, 10.77. Found: C, 58.51; H, 6.14; Br, 20.39; N, 10.86.

### 3.2. Molecular Modeling

Prior to molecular docking, the geometries of the compounds were quantum-mechanically (QM)-optimized using GAMESS-US [37] software (B3LYP/6-31G*). The optimized structures of the ligands were used with partial atomic charges derived from QM results [38,39,40,41] according to the Mulliken scheme [42] without torsion or any other degree of freedom for the ligands, which were frozen for molecular docking.

X-ray crystal structures of human AChE PDB ID:4EY4 [43] and human BChE PDB ID:1P0I [44] were used for molecular docking after molecular-mechanical optimization of side chains [45] and removal of water and other ligands.

Molecular docking (rigid protein, flexible ligand) with a Lamarckian genetic algorithm (LGA) [46] was performed using AutoDock 4.2.6 software [47]. The grid box for docking included the entire active site gorge of AChE (22.5 Å × 22.5 Å × 22.5 Å. grid box dimensions) and BChE (15 Å × 20.25 Å × 18 Å grid box dimensions) with a grid spacing of 0.375 Å. The main LGA parameters were 256 runs, 25 × 10^6^ evaluations, 27 × 10^4^ generations, and a population size of 3000. Docking with AutoDock Vina [48] 1.1.2 was performed with exhaustiveness of 40 and 1000.

### 3.3. Biological Studies

#### 3.3.1. In Vitro Cholinesterase Inhibition Assay

The inhibitory potency of compounds against hAChE and hBChE was evaluated using Ellman’s method [32]. Acetylthiocholine iodide, butyrylthiocholine iodide, recombinant human AChE, BChE from human plasma, and 5,5’-dithio-bis-(2-nitrobenzoic) acid (DTNB) were purchased from Sigma-Aldrich (Sigma-Aldrich, Germany). Stock solutions of compounds (0.01 M) were made using ethanol or water. In the case of compounds soluble in ethanol, the final concentration of ethanol in the cuvette was 1% vol. All assays were performed at 25 °C using a PerkinElmer λ25 spectrophotometer at 412 nm. The enzyme-catalyzed hydrolysis reaction was carried out in 0.1 M phosphate buffer at pH 8.0 containing 0.25 units of AChE or BChE and 1 mM acetylthiocholine or butyrylthiocholine as substrates. The tested compounds were pre-incubated with the enzyme for 5 min at 25 °C prior to adding the substrate and starting to record hydrolysis kinetics. Experiments were conducted in triplicate. The rate of substrate hydrolysis, measured by optical density change at 412 nm change over 2 min, was calculated. The sample without substrate was used as a blank. The sample without inhibitor was used as a control (in a solution of 1% vol. ethanol in the case of compounds soluble in ethanol). Percentage of AChE/BChE inhibition was determined by comparison of rates of reaction of test samples relative to the control sample. IC50 (the concentration of the drug producing 50% enzyme activity inhibition) was calculated by Hill plot using OriginPro 8.5 software.

#### 3.3.2. Acute Toxicity Evaluation and In Vivo BBB Permeation Assay

All experiments involving animals were performed in accordance with the guidelines set forth by the European Union Council Directive 2010/63/EU, approved by the ethical committee of the Kazan Medical University. Toxicological experiments were performed using intraperitoneal injection of the different compounds in mice weighing 20–25 g. Mice were maintained on a 12-h light/dark cycle (light from 7:00 a.m. to 7:00 p.m.) at 20−22 °C and 60%−70% relative humidity. Five different doses (determined during preliminary tests) were used with six animals per dose. Animals were observed 14 days after injection, and symptoms of intoxication were recorded. LD_50_ dose (in mg/kg) causing lethal effects in 50% of animals was taken as a criterion of toxicity. LD_50_ was determined by the method of Weiss [49]. Stock solutions of compounds were prepared in ethanol or water. Compounds were administered intraperitoneally in sterile physiological saline. The concentration of ethanol in physiological saline was 5% vol.

For in vivo BBB permeation assay the whole brains were removed 30 min after i.p. injection of the LD_50_ dose of the tested compound (experimental group, n = 6 mice) or after i.p. vehicle injection (control group, n = 6 mice). Brains were frozen in liquid nitrogen. Whole brain homogenates were prepared in a Potter homogenizer with 0.05 M Tris-HCl, 1% Triton X-100, 1 M NaCl, 2 mM ethylenediaminetetraacetic acid, pH 7.0, 4 °C, with 1 volume of brain with 2 volumes of buffer. The homogenate was centrifuged (10,000× *g* rpm, t = 4 °C) for 10 min using an Eppendorf 5430R centrifuge with a FA-45-30-11 rotor (Eppendorf AG, Hamburg, Germany). A 50 µL aliquot of supernatant was incubated with 5 µL of 0.5 mM tetra-isopropyl pyrophosphoramide (iso-OMPA) as a specific butyrylcholinesterase inhibitor, for 30 min. After that, the enzyme-catalyzed hydrolysis reaction was initiated by adding 10 µL of acetylthiocholine (0.01 M) as a substrate. After 10, 20, or 30 min of incubation with the substrate at 25 °C, the reaction was stopped by adding neostigmine (final concentration 0.01 M). Samples were diluted 25 times in 50 mM phosphate buffer (pH 8.0), and DTNB (0.1 mM) was added. Production of yellow 5-thio-2-nitro-benzoate anion, resulting from the reduction of DTNB by thiocholine (the product of enzymatic hydrolysis of acetylthiocholine), was measured spectrophotometrically according to the Ellman method [31]. The rate of thiocholine production over 20 min (from the 10th to the 30th min) was calculated. Brain samples of the control group were used as a control (100% of cholinesterase activity). The sample without substrate was used as a blank. All measurements for each brain sample were obtained in triplicate.

#### 3.3.3. Memory Performance Study: Scopolamine Model

Administration of scopolamine, a muscarinic acetylcholine receptor antagonist, produces a transient receptor blockade and cognitive deficits, which can be considered as AD modeling [50]. Mice were randomly divided into 6 groups (n = 9–11): Control group of mice; mice injected with a water solution of scopolamine (2 mg/kg, i.p.) at 40 min before testing; and mice treated with scopolamine (2 mg/kg, i.p.) and different dosages of compound **2b** (1, 5, and 10 mg/kg i.p.) dissolved in 0.1% water-based ethanol solution or donepezil (1 mg/kg, i.p.) at 40 and 20 min before testing, respectively. Drugs were administered over 18 days.

#### 3.3.4. Behavioral Tests

To evaluate spatial memory performance, mice were trained on a reward alternation task [51,52] using a conventional T-maze (OpenScience, Moscow, Russia). Before T-maze testing began, mice were placed on a food-deprivation schedule over 3 days and then given 4 days to habituate to the maze.

On each of the 14 training days, mice were given 6 pairs of training trials. In the first trial of each pair (forced trial), one of the goal arm guillotine doors was closed and the mouse was constrained to selecting the opposite arm. The mouse was returned to the start box 15–20 s after consuming the reward (diluted sweetened condensed milk). In the second (free-choice) trial, both goal arm doors were opened, but only the arm opposite the one selected in the forced trial was baited. The criterion for a mouse having learned the rewarded alternation task was 3 consecutive days of at least 5 correct responses out of the 6 free trials.

## 4. Conclusions

In conclusion, a series of AChE inhibitors based on 1,3-bis[ω-(substituted benzylethylamino)alkyl] uracil derivatives were synthesized. These compounds consisted of a uracil derivative moiety and polymethyleneamino-*ortho*-nitrilebenzyl chains at N1 and N3 atoms of the pyrimidine ring. For elucidation of structure–activity relationships of the novel nitriles, the polymethylene chains were varied from tetra- to hexamethylene chains, and secondary NH, tertiary ethylamino, and quaternary ammonium groups were introduced into the chains in addition to the use of 6-methyluracil and quinazoline-2,4-dione as uracil derivative moieties. In vitro experiments showed that the most active compounds, 1,3-bis[ω-(*ortho*-nitrilebenzylethylamino)butyl- and pentyl]-6-methyluracil, exhibited outstanding affinity against AChE with selectivity indexes for AChE over BChE exceeding 10,000. According to molecular modeling, these compounds bind AChE as dual binding site inhibitors. Theoretical and in vitro affinity towards AChE was supported by in vivo experiments, which showed that the most potent AChE inhibitor **2b** produced a therapeutic effect in cases of spatial memory deficit.

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
