# Peer review of "Novel Acetylcholinesterase Inhibitors Based on Uracil Moiety for Possible Treatment of Alzheimer Disease"

_molecules, 2020, doi:10.3390/molecules25184191_

Round 1

Reviewer 1 Report

I suggest the publication of the proposed manuscript in present form.

Author Response

The authors thank the reviewer for the high assessment of the article proposed in the journal Molecules.

Reviewer 2 Report

The manuscript entitled ‘Novel acetylcholinesterase inhibitors based on uracil moiety for possible treatment of Alzheimer’s disease’ by Vyacheslav E. Semenov, Irina V. Zueva, Marat A. Mukhamedyarov, Sofya V. Lushchekina, Elena O. Petukhova , Lilya M. Gubaidullina, Evgeniya S. Krylova, Lilya F. Saifina, Oksana A. Lenina, Konstantin A. Petrov, describes new series of uracil based compounds as strong and selective acetylcholinesterase inhibitors which bind to the peripheral anionic site and active site of the enyzme. In vitro and in vivo experiments defined the biological activity of compounds. Molecular docking studies revealed specific interactions of compounds and enzymes. This is very interesting and well written paper that could be of interest for medicinal chemistry readers. Manuscript is divided into sections which are all clear and well documented. Experimental design seems to be correct and data interpretations adequate. The title adequately reflects the subject of the manuscript and the summary give all necessary information. The manuscript includes up-to-date references. Before publication, the authors should correct the numbering of compounds because it is confusing (double numeration: (5a (2b . 2HBr) 5b (2c . 2HBr) 6 (4b . 2HBr), intermediates have higher numeration then resulting products thus the experimental section is not logically designed). In general, tables and figures should have more exact titles to be intelligible without reference to the text. My recommendation is to publish this article.

Reviewer 3 Report

Generally, the manuscript is well described and could be interesting for scientists working on new substances active against AD. Authors are focused on bifunctional inhibitors that simultaneously interact with both the catalytic site and the peripheral anionic site of AChE.

To me, however, some questions need explanation.

The solutions containing 5% of ethanol were administered intraperitoneally for acute toxicity studies. Is it possible to ignore the effect of ethanol alone?

It was found that a higher dose of compound 2b (10 mg/kg) produced less therapeutic effects in terms of spatial memory deficit. The results obtained for 10 mg/kg of 2b are similar to these of the scopolamine group. It needs more comments.

Scopolamine test allows exploration of some pathways in AD pathology. It lacks, however, hallmarks of AD, i.e. Aβ plaques and intraneuronal tangles of tau protein. Previous studies of the Authors [Ref. 23, 24] included experiments with  transgenic mice. This time there was no such research. Why?

“From the structure–activity profile of the compounds discussed, nitriles 2ac with the 6-methyluracil moiety and nitrile 6 with the quinazoline-2,4-dione moiety seem to be more promising for inhibition of AChE in vivo.”

There is no  any results from in vivo experiments for the compound 6.

Minor:

Scheme 1: K2CO?

Reviewer 4 Report

Several new uracil and quinazoline-2,4-dione derivatives were synthesized and tested against acetylcholinesterase (AChE) and butyrylcholinesterase (BChE). Some compounds showed potent inhibiting activity against AChE as well as moderate acute toxicity. The ability of synthesized compounds to cross the blood–brain barrier and inhibit brain AChE was demonstrated. One uracil derivative was shown to produce a therapeutic effect in cases of spatial memory deficit. The manuscript is interesting and can be accepted after following issues be addressed.

  1. 5a,b and 6 should be replaced by 2b x 2HBr, 2c x 2HBr and 4b x 2HBr correspondently. Fig. 2 should be deleted.
  2. Yields of all products should be provided in all chemical schemes.
  3. Two different compounds are designated as 5b (see Fig. 2 and Scheme 3).
  4. Was the death of animals in acute toxicity test caused by seizures similar to nicotine toxicity test?
  5. P. 5. "data for compounds 1a–f were taken from reference 25" but "the table also shows the data for inhibitors 1a–c and 1d–f as previously published [23]." Which reference is correct?
  6. "although selectivity of 1a–f for AChE vs. BChE was higher than that of 2a–c and donepezil hydrochloride". It is not so for 2b.
  7. "The acute toxicity of nitriles 2a–c was significantly less than the toxicity of nitro- and trifluoromethylcompounds and the reference drug." What about 1a?
  8. P. 6. "for nitriles 4a,c". There is no compound 4c in the manuscript.
  9. Pleas briefly discuss biological methods in the part "Behavioral tests" (crossing BBB and T-maze).
  10. "dynamic of reaching the criterion of learning the task" should be discussed.
  11. Conclusion. Please describe what do you mean by the word "selectivity".

Reviewer 5 Report

This is a very interesting and well conducted study describing novel acetylcholinesterase inhibitors based on uracil moiety for possible treatment of Alzheimer's disease

The study contains in vitro and in vivo study and has solid scientific background , is original and the results are well presented and discussed.

I have few minor suggestions:

For search of completeness in the introduction the Authors should recognize that several other disorders, life style factors and conditions leading to upregulated production of proinflammatory cytokines   may favour AD 

  Ko Y, Chye SM.Lifestyle intervention to prevent Alzheimer's disease.Rev Neurosci. 2020 Aug 17:/j/revneuro.ahead-of-print/revneuro-2020-0072/revneuro-2020-0072.xml. doi: 10.1515/revneuro-2020-0072. Online ahead of print.

Dafsari FS, Jessen F.Depression-an underrecognized target for prevention of dementia in Alzheimer's disease.Transl Psychiatry. 2020 May 20;10(1):160. doi: 10.1038/s41398-020-0839-1. Yu JT, Xu W, Tan CC, Andrieu S, Suckling J, Evangelou E, Pan A, Zhang C, Jia J, Feng L, Kua EH, Wang YJ, Wang HF, Tan MS, Li JQ, Hou XH, Wan Y, Tan L,   Mok V, Tan L, Dong Q, Touchon J, Gauthier S, Aisen PS, Vellas B. Evidence-based prevention of Alzheimer's disease: systematic review and meta-analysis of 243 observational prospective studies and 153 randomised controlled trials.J Neurol Neurosurg Psychiatry. 2020 Jul 20:jnnp-2019-321913. doi: 10.1136/jnnp-2019-321913. Online ahead of print.   Petralia MC, .The Role of Macrophage Migration Inhibitory Factor in Alzheimer's Disease: Conventionally Pathogenetic or Unconventionally Protective? Molecules. 2020 Jan 10;25(2):291. doi: 10.3390/molecules25020291.     Petralia MC, Mazzon E, Fagone P, Basile MS, Lenzo V, Quattropani MC, Bendtzen K, Nicoletti F.Pathogenic contribution of the Macrophage migration inhibitory factor family to major depressive disorder and emerging tailored therapeutic approaches.J Affect Disord. 2020 Feb 15;263:15-24. doi: 10.1016/j.jad.2019.11.127. Epub 2019 Nov 30     Petralia MC, Mazzon E, Fagone P, Basile MS, Lenzo V, Quattropani MC, Di Nuovo S, Bendtzen K, Nicoletti F.The cytokine network in the pathogenesis of major depressive disorder. Close to translation? Autoimmun Rev. 2020 May;19(5):102504. doi: 10.1016/j.autrev.2020.102504. Epub 2020 Mar 13.

Round 2

Reviewer 3 Report

My suggestions were taken into account. I think that the article can be published in Molecules.

This manuscript is a resubmission of an earlier submission. The following is a list of the peer review reports and author responses from that submission.

Round 1

Reviewer 1 Report

The manuscript is interesting and there are some promissing results.

However, there is to much information in the introduction. One third of this part is a description of previous papers of Authors, given on the Fig.1. I suggest to leave on Fig.1. only the most active derivatives, published before, without numbering the first of them as 1. I is a known compound, but it looks like a new. Are these 1 the same as 1a-1f? The description of substituents in Table 1 suggests they aren't.

(Fig.1. - Add citations of previously reported compounds)

There is not a clear statements, what is the link between molecular modelling, in vitro and in vivo tests? In the manuscript all these parts seems to are separated and not linked to each other. Write a substantive conclusion.

Reviewer 2 Report

This manuscript could be very interesting for scientists interested in the field of Alzheimer's disease, however I cannot recommend the publication because this manuscript lacks many requirement as it is.

If the authors wish to publish they will need to ask someone to check the language (example lines 33 or 43). Written English is not the only problem and here are some comments for the authors in order to help them to improve (and hopefully publish) this work. To this effect, I would recommend another medicinal chemistry oriented journal, if the molecular modelling is completely re-written and explained.

I regret very much to refuse this otherwise interesting piece of work, but molecular modelling and docking are not accessories to “improve” a scientific article, nor is it a way to get nice pictures. In the submitted paper, docking is used to justify the higher activity of some compounds with completely wrong experiments. Nobody does rigid ligand docking anymore, these days. If one did, I would ask why and would expect a full account of their arguments. There are none in the paper, while a software allowing full ligand flexibility was used, and even permit some flexibility in the active site… It is a pity that a scientific paper with synthesis, in vitro testing and animal testing presented such a poor quality in the field of modelling.

Below are more comments to improve this work.

Lines45-47, the sentence starts with “currently” and refers to a more than 15 years old paper.

Lines 67-70, the authors should explicite how this was demonstrated since one will not wish to read their whole bibliography to know. By the way, I have never found 12 references out of 48 belonging to the submitting group. Auto-citation should be much more limited.

Lines 84-85, should not the authors explain why these chain length ? Is there a rationale or shall we conclude it’s purely random ?

Lines 87-89, this is wrong. I have never seen someone writing or explaining this in 20 years in the field of molecular modelling. When one perform docking, one suppose that physiological pH is 7.4, which means amines are protonated and carboxylates are unprotonated. Local environment can influence this, and a lot of talk had been debated, a long time ago on the subject. Indeed, the solubility of the compounds will change and this might influence the testing outcome, but not because of the added HBr… How did the authors check the presence of Br counter anion ?

When presenting docking results, write what you are talking about, a table with energies or scores is necessary and argument about whether you use the first obtained pose or the representative clusterized one…

All the schemes suffer a common problem: drawings are of poor quality, angles, bond length, some lines look like bold ones, colors are not necessary since purple is not explained (CN group).

Footnotes of scheme 1 and figure 1 present errors. Scheme 2 is not necessary, only one scheme should suffice. The drawn molecules should be of the same size all over the different schemes and figures.

Line 136, C-35 refers to something not explained in this article.

Line 137, when referring to an active site, it would be a good idea to explain a bit for the reader. What is this area composed of ? A picture with colors can be of help… Never present something without explanation or arguments.

Lines 139-143, please cite a scientific peer-reviewed paper where someone did this ? You can do what you wish with docking and the submitted structures, it does not mean this is right. In silico always gives a result. By the way, the description sentence is really too long and unhelpful without a corresponding picture to explain.

Figure 3 and should not present two different sizes of molecules, but only one.

Results in Table 1 are not sufficiently different to claim what is written from lines 189-208. A 10 times change might be a difference. In the footnotes, the authors refers to reference 25, which means that some results in the table are more than 5 years old ? Are you serious about this, comparing old experiments to new one when this should be checked again for compound 1a-f ? When one can perform behavioural tests on mice, one can redo some simple enzymatic experiments.

The molecular modelling (choose this orthography or modeling, but not both) is really poor, how could someone redo your experiments ? Why such complicated way to calculate a simple point charge, when performing fully rigid docking ? How were the proteins and ligands prepared ?

The most important thing to publish this work will be to perform or have perform correct modelling on the compounds. Vina software is really fast and uses the same requisites as autodock, that’s a tip. Without it, such virtual work should not be published.

Reviewer 3 Report

The manuscript presented by Petrov and coworkers is based on the design, synthesis and both in vitro and in vivo characterization of some new uracil based Achetylcholinesterase inhibitors. According to the results obtained by molecular modeling studies, the compounds act as dual inhibitors binding to both the peripheral anionic site and the active site of the target. The authors idea draws its origin from the results published on the same topic in 2015. In particular the starting point for the authors was a series of 6-methyluracil derivatives with ω-(substituted benzylethylamino)alkyl chains at the nitrogen atoms of the pyrimidine ring. The structural changes in this novel series of compounds concern the numbers of methylene groups in the alkyl chain and the introduction of a new substituent on the benzyl ring, as reported in their previous work. Indeed, the authors discuss the in vitro inhibitory activity of the new compounds comparing to that registered for the previously characterized compounds. In order to make the discussion of the obtained results more consistent I suggest the authors to perform the following revisions in order to make the manuscript published.

According to the data reported in table 1 it seems quite clear that the best results for each series of compounds are obtained for compounds showing a number of methylene groups equal to 3 in the alkyl chain. Looking at the results from this new point of view the most active compounds are 1b and 1e followed by compound 2b and 4b. The author might discuss this aspect and eventually confirm this trend also by molecular modeling study.

As comment to the results reported in table 1 the authors declare: “Quaternization of atoms of N in nitriles 2b,c with HBr acid, compounds 5a,b, gave a slight increase of inhibitory power against AChE, decrease of selectivity and significant decrease of LD50. Contrary, quinazoline-2,4-dione 6 with quaternized N atoms in pentamethylene chains compared to neutral counterpart 4b demonstrated almost the same inhibitory activity against enzymes studied but significantly lower acute toxicity. “ How can the authors explain this aspect? The authors should be able to explain why compounds 5a, b showed a higher toxicity profile than the corresponding compounds 2b, c. In my opinion these results might be ascribed to the different solubility of the species. I suggest the authors to perform some solubility tests in order to evaluate the solubility of all these compounds.

For what concern the results reported in figure 6 on the in vivo tests, the significance level and the error bars might be provided for the results obtained at each concentration of compound 2b in both the correct choice and task learned graph.

Please make figure1 more consistent with what is reported along the text. The description of compounds structure reported in figure 1 is not clear. The authors have to assign a number to each structure. They may refer to compounds which only differ in alkyl chain length, as 1a 1b 1c. For derivatives of compounds 2 and 3 the number of alkyl moieties is not showed in Figure 1. Please report n as equal to 2-4.  According to the chain length compounds have to be named 2a 2b 2c as well as 3a 3b and 3c.

The structural modifications reported for compound 3 and 4 have to be explained in a clearer and more consistent way.

Moreover for compounds belonging to group 1, six substituents on benzyl group are described! This sounds quite strange!!!!

Line 129 correct hydrogalogenation as hydrohalogenation.

The dosages for behavioral test have to be reported in the experimental section too.

The english language has to be extensively revised.

Round 2

Reviewer 1 Report

The article can be published in the presented form.

Reviewer 2 Report

The manuscript was much improved but both the docking section in the manuscript and the authors anwsers are not sufficient to explain the modeling described to readers, nor relevant to explain the biological results.

Reviewer 3 Report

Despite the attempts to improve their manuscript, the authors missed to solve a crucial aspect of their work. The results reported in figure 6 still remain of questionable significance. Indeed, a p>0.05 characterizes the results obtained after treatment with Donepezil in both the correction choice and task learned experiment. That makes the reported results negligible.

The correlation between compounds aqueous solubility and toxicity has to be reported in the discussion.

The mass spectra for compound 2b reveals the presence of compound 2a. Then, the IC50 value reported for this compound might be different. I suggest the author to carry out the inhibition experiment by using the purified compound.  

The authors declare the activity of compounds 2a-c as well as 1a-c and 1d-f decreases with the increase of chain lenght. This statement is not in agreement with the reported IC50 values that clearly shows the compounds with a chain lenght of 5 methylene units to be the most active toward AChE. This aspect is also discussed in the new version of the manuscript in lines 167-171.

Compound 2c is not mentioned in the caption for Figure 1.

In table 1 is reported the in vitro activity for compound 13. On the contrary the authors claim, both in the caption and along the text, to report the activity for compound 9 in the same table.